# Seroprevalence of Four Polyomaviruses Linked to Dermatological Diseases: New Findings and a Comprehensive Analysis

**DOI:** 10.3390/v14102282

**Published:** 2022-10-17

**Authors:** Krisztina Jeles, Melinda Katona, Eszter Csoma

**Affiliations:** 1Doctoral School of Pharmaceutical Sciences, University of Debrecen, 4032 Debrecen, Hungary; 2Department of Medical Microbiology, Faculty of Medicine, University of Debrecen, Nagyerdei krt. 98., 4032 Debrecen, Hungary

**Keywords:** MCPyV, HPyV6, HPyV7, TSPyV, seroprevalence

## Abstract

Our aim was to study the seroprevalence of human polyomaviruses (HPyV) linked to skin diseases. A total of 552 serum samples were analysed by the enzyme-linked immunosorbent assay to detect IgG antibodies against Merkel cell polyomavirus (MCPyV), HPyV6, HPyV7 and Trichodysplasia spinulosa-associated polyomavirus (TSPyV) using recombinant major capsid proteins of these viruses. The individuals (age 0.8–85 years, median 33) were sorted into seven age groups: <6, 6–10, 10–14, 14–21, 21–40, 40–60 and >60 years. The adulthood seroprevalence was 69.3%, 87.7%, 83.8% and 85% for MCPyV, HPyV6, HPyV7 and TSPyV, respectively. For all four polyomaviruses, there was increasing seropositivity with age until reaching the adulthood level. There was a significant increase in seroreactivity for those age groups in which the rate of already-infected individuals also showed significant differences. The adulthood seropositvity was relatively stable with ageing, except for TSPyV, for which elevated seropositivity was observed for the elderly (>60 years) age group. Since seroepidemiological data have been published with wide ranges for all the viruses studied, we performed a comprehensive analysis comparing the published age-specific seropositivities to our data. Although the cohorts, methods and even the antigens were variable among the studies, there were similar results for all studied polyomaviruses. For MCPyV, geographically distinct genotypes might exist, which might also result in the differences in the seroprevalence data. Additional studies with comparable study groups and methods are required to clarify whether there are geographical differences.

## 1. Introduction

The first two human pathogenic members of the *Polyomaviridae* family, BK and JC polyomaviruses (BKPyV and JCPyV, respectively), were discovered in 1971 [1,2]. Out of the 13 new members that have been described from human samples since 2007, 11 are classified as human polyomaviruses (HPyVs) by the International Committee of Taxonomy of Viruses: Karolinska Institute PyV, Washington University PyV, Merkel cell PyV (MCPyV), HPyV6, HPyV7, Trichodysplasia spinulosa (TS)-associated PyV (TSPyV), HPyV9, HPyV10, Saint Louis PyV, New Jersey PyV and Lyon IARC PyV [3,4]. Besides BKPyV and JCPyV, only four novel members of the family, MCPyV, HPyV6, HPyV7 and TSPyV, have been linked to human diseases, all of which are related to the skin [5]. The transmission routes have not been clarified for any of the HPyVs, but infection of the skin may occur for the four above-mentioned viruses, resulting in persistent/latent infection. These infections are thought to be asymptomatic in the vast majority of patients, but they can result in severe consequences, mainly in immunocompromised patients [6].

MCPyV was described from a very aggressive, neuroendocrine tumour of the skin, Merkel cell carcinoma (MCC) [7]. MCC is a rare cancer with an overall incidence of 0.1–1.6/100,000 people per year; however, a continuous increase in the number of the cases has been reported [8,9,10,11]. MCC mainly occurs in fair-skinned and elderly patients, and exposure to ultraviolet (UV) radiation and immunosuppression are key risk factors. Unfortunately, the mortality rate of MCC is high and exceeds that of melanoma [12]. As a clonally integrated tumour virus, MCPyV is a main aetiological agent for MCC; based on meta-analyses, the overall prevalence of the virus is 80% in MCC cases. There is a geographical difference in prevalence rates of MCPyV-positive MCC, with the lowest rate in Australia (18–24%), and MCC without MCPyV also exists [5,13,14]. Besides MCC, MCPyV DNA has been detected in healthy skin, non-MCC skin lesions [13], blood, urine, respiratory specimens, gastrointestinal and lymphoid tissue samples [14,15]. Based on seroepidemiological studies, MCPyV is prevalent in the human population, establishing a lifelong, mostly asymptomatic infection in healthy individuals. The seropositivity in the healthy population increases with age from early childhood, but the published adulthood seroprevalence varies widely, ranging from 46% to 87% [16].

In 2010, HPyV6 and HPyV7 were described in healthy human skin [17]. The infections are prominently asymptomatic in immunocompetent individuals, but HPyV6 and HPyV7 are the causative agents of pruritic and dyskeratotic dermatosis in immunocompromised patients [18,19]. Both viruses are frequently detected in healthy skin specimens [17,20,21,22,23,24,25], but they have also been found in urine, blood, lymphoid tissue and respiratory specimens. HPyV6 and HPyV7 are ubiquitous viruses, but similarly to MCPyV, the published seroprevalence in adults varies widely: 52–93% for HPyV6 and 33–84% for HPyV [26]. Although the oncogenic potential of HPyV6 and HPyV7 has been hypothesised, and the prevalence of both viruses has been studied in skin and non-cutaneous malignancies, no evidence has been found to prove their role in any oncogenesis [26].

TSPyV was discovered from TS, a rare skin disease characterised by folliculocentric papules with keratin spikes, and it exclusively affects severely immunocompromised patients [27]. The virus has been detected in different sample types, such as skin, urine, blood, respiratory, lymphoid and gastrointestinal, but it has only been proven as the causative agent of TS [28,29]. TSPyV is a widespread infection: the seropositvity of the healthy, adult population is 63–80%, and primary infection seems to occur mainly during childhood [28]. Although BKPyV and JCPyV result in severe clinical consequences due to the reactivation of latent viruses rather than primary infection, it has been proven that TS can be caused by primary TSPyV infection. This and the high rate of primary infection during childhood might explain why TS is a very rare disease among immunocompromised patients, and it highlights the importance of serological studies [30].

Whether primary infection or reactivation causes a disease, it is necessary to know the prevalence of these polyomaviruses in the population to assess the risk of a possible disease or even oncogenesis. There may be several explanations for the differences in published seroprevalence data. One very obvious reason is different methodologies, including the immunoassay and the protein used as an antigen. Researchers have suggested that there are geographically distinct genotypes of MCPyV [31,32]; however, there is no evidence for the existence of serotypes. Differences among the cohorts (age and clinical status) might result in different data.

In this study, we determined the seroprevalence of four polyomaviruses—MCPyV, HPyV6, HPyV7 and TSPyV—linked to skin diseases. To detect the virus-specific antibodies in serum specimens, we developed and optimised indirect enzyme-linked immunosorbent assays (ELISAs) for which the virus-specific antigens were expressed in bacteria. Since the published seroepidemiological data vary widely for the viruses studied, we performed a comprehensive analysis with data available from the literature. We compared published age-specific seropositivities with our data.

## 2. Materials and Methods

The Regional and Institutional Research Ethics Committee, Clinical Centre, University of Debrecen, Hungary, approved the study (DE RKEB/IKEB: 5134-2018). Written consent was not required from the patients because serum samples were used retrospectively and anonymously.

### 2.1. Samples and Patients

A total of 552 serum samples sent for routine serological diagnostic tests (herpes, hepatitis and coronavirus) to Medical Microbiology, University of Debrecen, Hungary, were analysed. The sera were collected between 2016 and 2021 and stored at −70 °C until testing. The study population ranged from 0.8 to 85 years (median 33 years) and 283 females and 269 males were included. The samples were divided into the following age groups: <6 years (n = 38), 6–10 years (n = 36), 10–14 years (n = 45), 14–21 years (n = 87), 21–40 years (n = 114), 40–60 years (n = 128) and >60 years (n = 104). Table 1 shows the patient data by age groups.

### 2.2. Antigen Production for ELISA

MCPyV, HPyV6, HPyV7 and TSPyV major capsid proteins (VP1) were produced as detailed previously [33]. Briefly, codon-optimised VP1 genes of HPyV6 (GenBank accession number: NC_014406), HPyV7 (GenBank accession number: NC_014407), MCPyV (GenBank accession number: NC_010277) and TSPyV (GenBank accession number: NC_014361) were inserted into the pTriEx™-4 Neo vector (Novagen, Pretoria, South Africa; Merck, Kenilworth, NJ, USA). Protein expression was carried out in Origami™ B(DE3)pLacI competent cells (Novagen). Protino Ni-TED Packed Columns (Macherey-Nagel, Düren, Germany) was used to purify the viral proteins from the inclusion bodies. Following dialysis and concentration, the viral proteins were analysed qualitatively and quantitatively. Coomassie brilliant blue staining of the proteins is shown in Figure 1. The Pierce BCA Protein Assay kit (Thermo Fisher Scientific, Waltham, MA, USA) was used for quantitative measurements. The identity of the VP1 proteins was confirmed by Western blotting.

### 2.3. Detection of Anti-Polyomavirus Antibodies by ELISA

The details of our in-house ELISA are available in our previous publication [33]. The antibodies against the polyomavirus major capsid proteins were detected in duplicates for each serum sample. The optical density (OD) values were determined as the average after subtraction of the blank value. The cut-off value was determined for each ELISA based on the inflection point of the graph obtained from the function of the tendency curve of the ranked OD values plotted. Based on this, samples were considered seroreactive if the OD values were >0.146, >0.222, >0.265 and >0.375 for MCPyV, HPyV6, HPyV7 and TSPyV, respectively.

An antigen-competition assay was carried out with the same ELISA protocol, but serum samples were pre-incubated with 1000 ng (20-fold excess amount) of homologous or heterologous VP1 proteins.

### 2.4. Statistical Analysis

GraphPad Prism version 9.4.0 (GraphPad Software, San Diego, CA, USA) was used for the following statistical analyses: chi-square for trend, Fisher’s exact test, Mann–Whitney test and Spearman’s rank correlation.

## 3. Results and Discussion

### 3.1. Seroresponses

We measured seroresponses against MCPyV, HPyV6, HPyV7 and TSPyV with ELISA, using VP1 proteins of the viruses as antigens. The obtained optical density values by age groups are shown separately for each virus in Figure 2. We used the Mann–Whitney test for pairwise comparison of the OD values among the age groups. The MCPyV seroresponse was not significantly different among the age groups (Figure 2a). For HPyV6, there was a significant difference between the <6 years and 6–10 years age groups (*p* = 0.0359) and between the 14–21 and 21–40 years age groups (*p* = 0.0192) (Figure 2b). For both comparisons, the older age group showed higher mean and median OD values. For the HPyV7 ELISA, there were significantly different seroresponses between the 10–14 years and 14–21 years age groups (*p* = 0.021) and between the 14–21 years and 21–40 years age groups (*p* = 0.003). The mean and median OD values were higher for the older age groups (Figure 2c). For TSPyV, there were significant differences between the <6 years and 6–10 years age groups (p = 0.009) and between the 40–60 years and >60 years age groups (*p* = 0.029) (Figure 2d).

We performed correlation analysis to determine the potential association between the seroresponses against the polyomaviruses measured in different ELISAs (Figure 3).

There was a moderate correlation between the HPyV6 and HPyV7 OD values (*r* = 0.326, *p* < 0.0001). These viruses are closely related and belong to the same genus (*Deltapolyomavirus*) of the *Polyomaviridae* family [4]. The amino acid sequences of their VP1 antigens have 69% identity (according to the Basic Local Alignment Search Tool (BLAST), NCBI). There was little or no correlation for the other ELISA results. The correlation coefficient for MCPyV and TSPyV ELISA was the second highest, but relatively low (*r* = 0.277, *p* < 0.0001) (Figure 3). These two viruses are closely related, belonging to the *Alphapolyomavirus* genus [4]. Based on BLAST analysis, the sequence identity of the major capsid proteins used for MCPyV and TSPyV ELISA is 57%. Despite the significant but weak correlations of the OD values between HPyV6 and HPyV7 and between MCPyV and TSPyV, the seroresponses are thought to be specific for the given polyomavirus. Pre-incubation of the sera with the heterologous antigen did not significantly change the OD values, while the homologous VP1 neutralises the antigen–antibody reaction). Our data are in accordance with other publications [34,35,36,37].

### 3.2. Seroprevalence of MCPyV, HPyV6, HPyV7 and TSPyV

We calculated seropositivity for each polyomavirus as the proportion of the samples with an OD value above the determined cut-off. The overall seropositivity was 63.9%, 79.2%, 72.5% and 78.4% for MCPyV, HPyV6, HPyV7 and TSPyV, respectively, while the adulthood (>18 years) seroprevalence was 69.3%, 87.7%, 83.8% and 85% for MCPyV, HPyV6, HpyV7 and TSPyV, respectively.

Age-specific seroprevalences of the studied polyomaviruses are shown in Figure 4. The seropositivity for all four polyomaviruses increased significantly with age (Χ^2^ test for trend; *p* = 0.0004 for MCPyV and *p* < 0.0001 for HPyV6, HPyV7 and TSPyV). This finding is consistent with published data [26,38,39,40]. Our data strengthen the findings that primary infection occurs in early childhood: the vast majority is possible in children and young adults, but it may happen throughout life. There was a significant increase in seroprevalence in different age groups for the studied polyomaviruses. For MCPyV (Figure 4a), there was a significant increase between 10–14 and 14–21 years; for HPyV6 (Figure 4b) and HPyV7 (Figure 4c), there was a significant increase between 14–21 and 21–40 years. The significant and highest increase in seropositivity for TSPyV was in the 6–10 years age group. These findings are in agreement with the results of seroreactivity analysis: the age groups with significantly higher seropositivity showed significantly higher OD values for HPyV6, HPyV7 and TSPyV (Figure 2b–d and Figure 4b–d). After reaching the adulthood level of seroprevalence in the 14–21 years age group (for MCPyV and TSPyV) or the 21–40 years age group (for HPyV6 and HPyV7), the antibody levels against the viruses remained relatively stable. There was a significant increase in seroprevalence in the oldest age group (>60 years) for TSPyV. Increased seropositivity in elderly patient groups has been observed by others [34,36]. This increase may be due to the fact that primary infection can occur at any time during life, resulting in increasing seroprevalence with age. However, similarly to the findings of Šroller et al. [36], the seroprevalence among the adults < 60 years in our study cohort was relatively stable, and then increased (Figure 4d). This phenomenon might be the consequence of reactivation of latent infection.

Although the proportion of female and male individuals in our study cohort was not markedly different (51.3% vs. 48.7%), there were significantly higher seropositivity rates among women for MCPyV (195/283 vs. 158/269; *p* = 0.0133), HPyV6 (234/283 vs. 203/269; *p* = 0.0461) and HPyV7 (218/283 vs. 182/269; *p* = 0.017). This significantly higher female seroprevalence occurred among children (<18 years) but not among adults (≥18 years), and only for HPyV6 (68/96 vs. 54/97; *p* = 0.0366) and HPyV7 (59/96 vs. 40/97; *p* = 0.0062) (Table 2). Additional investigation with a larger study group is required to examine whether this phenomenon is specific for that patient group.

In the literature, there is a wide range of the overall seroprevalence of the four polyomaviruses we evaluated. There are several reasons for the variability in published seroprevalence data, including geographical differences and differences in the methods used, the VP1 proteins used as antigens and the study cohorts. To evaluate our findings, we performed a comprehensive data analysis comparing our seropositivity data to those published by others. We performed a pairwise comparison (Fisher’s exact test) of data between age groups and cohorts for the four studied polyomaviruses. The results are shown separately for MCPyV (Figure 5), HPyV6 (Figure 6), HPyV7 (Figure 6) and TSPyV (Figure 7). Each figure represents a comparison of our data with data from a previous publication. We sorted our samples into age groups that are comparable to the published studies. The seropositivity by age groups, significant differences, the geographic location of the samples, the total number of samples and the isolate used as an antigen for MCPyV only (if published) are shown for each figure. Detailed patient data (such as age and serostatus of each individual) are not available in most of the publications, so even if the mean and/or median ages have been published for the study groups, there might be differences, even significant differences. This might be a limitation of these analyses, because there is an obvious correlation between the seropositivity rate and age.

After sorting our samples into comparable age groups, we found that despite the differences in methods used, we have very similar, statistically insignificant overall MCPyV seroprevalence compared with data from the Czech Republic [41], Italy [16,42], Iran [43], Cameroon [44] and Australia [35] (Figure 5a–f). In fact, there were no significant differences in the age group-specific seropositivities for most of the groups of the cohorts compared. The few significant differences were for children and young adults. Seroprevalence increased with age until adulthood for all cohorts, but the slope of this trend could be different, meaning that the vast majority of the primary infections during childhood may occur at a variable age. There have been significantly higher overall MCPyV seroprevalences published for Spain [40,45,46], the Netherlands [38] and Italy [47,48] (Figure 5g–l). The MCPyV isolates used as antigens, the protein expression systems and the immunoassay methods are variable among these studies. When considering individuals < 19 years old, our seroprevalence was not different from that reported by Viscidi et al. [47], and the adulthood seroprevalence was very similar until 50 years of age. The other authors did not publish comparable childhood data [38,40,45,46,48]. There were significantly lower overall seropositivities from Japan [49], China [50] and the USA [51,52] (Figure 5m–p). Pairwise comparison also revealed significant differences between most of the age groups for two studies [49,51]. Tolstov et al. [39] also detected markedly (but not significantly) lower adulthood seropositivity in sera from the USA compared with our data (Figure 5q). For some publications, we were unable to perform pairwise comparisons for age groups due to the absence of detailed data or because the study cohorts were not comparable. In summary, we observed very similar or somewhat or significantly different MCPyV seropositivity in our study cohort compared with studies published by other researchers using similar or different methods. Usage of virus-like particle (VLP) composed of VP1 proteins or a tagged recombinant VP1 (e.g., glutathione-S-transferase (GST)-fused VP1) protein as antigens may result in differences in an immunoassay [36]. At the same time, our MCPyV seroprevalence obtained from an ELISA using a tagged VP1 protein produced very similar results as observed by others using VLP [41,44] or the peptide part of VP1 [16] as an antigen. Some of the research teams used different immunoassay techniques, such as colorimetric ELISA [16,39,41,43,44,45,46,47,48,51] [and the Luminex bead-based assay [35,38,40,42,49,50,52], and they found very similar or significantly different results using these different methods. The VP1 proteins used as antigens varied among the examinations; for example, major capsid proteins of different MCPyV isolates were used. The existence of geographically related genotypes has been suggested based on phylogenetic analysis [31]. Li et al. [49] used the VP1 protein of the MCPyV TKS isolate, which originates from Japan, and classified it into a separate clade relative to the clade into which most of the isolates used by others and ourselves belong. It is hypothesised that the differences in the seroprevalence are due to geographical differences. To clarify this, several sequences from different geographical regions should be collected and analysed—to identify genotypes—and seroprevalence studies should be performed using different genotypes, specific to or different from a given geographical region.

The overall and adulthood HPyV6 (79.2% and 87.7%) and HPyV7 (72.5% and 83.8%) seroprevalence in our study cohort was also within the range published by others (52–93% for HPyV6 and 33–84% for HPyV7) (Figure 6) [26]. The adulthood HPyV6 and HPyV7 seroprevalence from the Czech Republic is similar to ours. Those authors carried out ELISA-based measurements using both VLP and GST-VP1 as antigens. The only significant difference was a lower seroprevalence for the young adults compared with our age groups (Figure 6a,b) [36]. The Dutch cohort showed a similar overall HPyV6 infection rate with the exception of significantly lower data in the 18–29 years age group (Figure 6c), but HPyV7 seroprevalence was significantly lower compared with what we detected (Figure 6d). They used the fluorescent bead-based immunoassay with GST-VP1 [38]. The adulthood seropositivity of people > 40 years from Spain was significantly higher for HPyV6, but for HPyV7 it was similar to what we observed (Figure 6e,f). Their method was a fluorescent bead-based assay using GST-VP1 antigen [40]. Using that same method, authors from Australia found similar seropositivity rates within the youngest age groups (<14 years), but adulthood and total seroprevalence were significantly lower for both HPyV6 and HPyV7 compared with our data (Figure 6g,h). Seropositivity increased with age in both study groups, but while we found a relatively stable infection rate over 40 years, they also observed an increasing trend for adults [35]. The adulthood (>20 years) HPyV6 seroprevalence (83.4%) was very similar in the Italian cohort to in our study group (87.7%) [34]. They performed VLP-based ELISA. Schowalter et al. [17] reported low HPyV6 and HPyV7 seropositivity (69% and 35%) in the USA using VLP-based ELISA in a pilot study with 95 serum samples. We could not perform pairwise comparison for these studies. Despite the variability in the methods, seroprevalence data for HPyV6 and HPyV7 are mostly similar; however, there were differences in overall seropositivities and in increasing trends. This variability might be due to the different study groups, or even geographical differences.

The adulthood TSPyV seroprevalence (85%) from this study was slightly higher compared with previously reported rates of 63–80% (Figure 7) [28]. Several studies reported significantly lower total seroprevalence [34,35,36,53,54,55,56]. Despite the differences in total seroprevalence, we observed similarities in the data. In the Australian cohort, the increasing trend with age was very similar to what we observed: the seroprevalence for individuals < 10 years [35,53] and > 40 years [35] and the adulthood (>18 years) seropositivity rates were not significantly different (Figure 7a,b) [35]. Seropositivity also increased with age in the Japanese cohort (Figure 7c). Compared with our data, there was no significant difference in seropositivity in the 30–69 years age groups [54]. As we observed, there was also an increase in antibody positivity with ageing (>59 years) reported for the Czech Republic (Figure 7d) [36] and Australia (Figure 7a) [35]. At the same time, there was decreased seropositivity in the elderly group (>70 years) in the Spanish (Figure 7e) and Japanese (Figure 7c) cohorts [40,54]. There was no significantly different adulthood seroprevalence reported by Kamminga et al. [38] from the Netherlands (Figure 7f) and Gossai et al. from the USA (80.9%,) [57].

## 4. Conclusions

We observed similar seroprevalence for MCPyV, HPyV6, HPyV7 and TSPyV to those published by other research groups. At the same time, the available seroprevalence data cover a wide range for each virus and differences can be observed not only in the overall data, but also in the trends regarding how seropositivity changes with age. Although the cohorts, methods and even the antigens vary among the studies, there are similar results. For MCPyV, geographically distinct genotypes might exist that might underlie differences in the reported seroprevalence. Additional studies with comparable study groups and methods are required to clarify whether differences occur geographically.

## Figures and Tables

**Figure 1 viruses-14-02282-f001:**
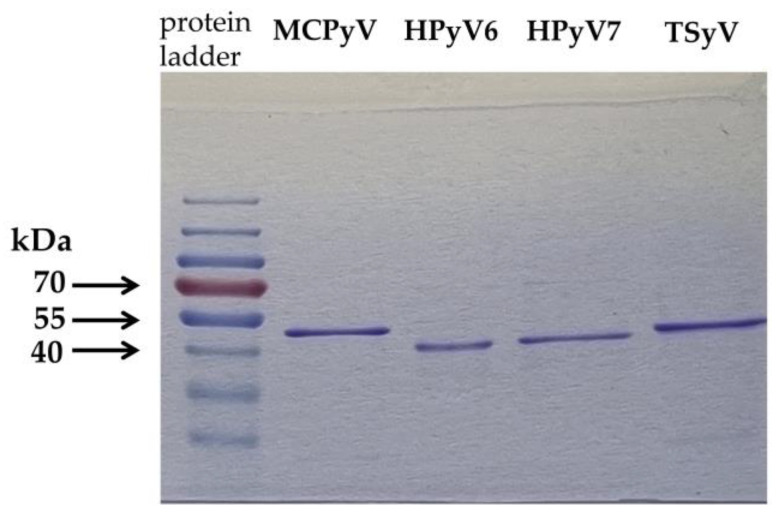
Analysis of the MCPyV, HPyV6, HPyV7 and TSPyV major capsid protein production using Coomassie brilliant blue-stained SDS-PAGE (sodium dodecyl sulphate–polyacrylamide gel electrophoresis). In lanes (from left to the right): 1, PageRuler Pre-stained Protein Ladder, 10–180 kDa (Thermo Fisher Scientific, Waltham, MA, USA); 2, MCPyV (Merkel cell polyomavirus) major capsid protein; 3, HPyV6 (Human polyomavirus 6) major capsid protein; 3, HPyV7 (Human polyomavirus 6) major capsid protein; 4, TSPyV (Trichodysplasia spinulosa-associated polyomavirus) major capsid protein.

**Figure 2 viruses-14-02282-f002:**
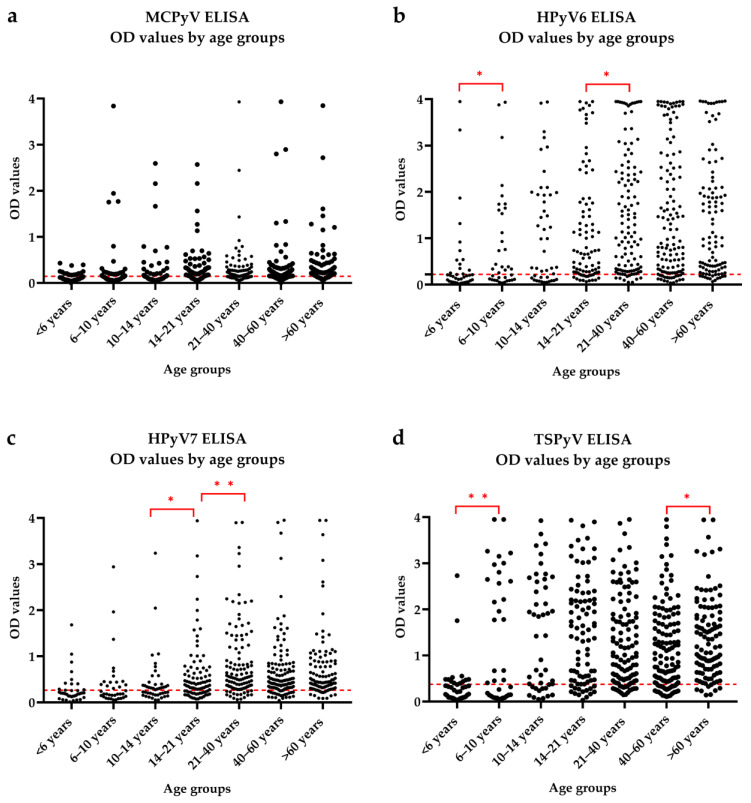
Age distribution of seroresponses against the major capsid proteins of MCPyV (**a**), HPyV6 (**b**), HPyV7 (**c**) and TSPyV (**d**). Each dot represents the optical density (OD) value of an individual serum sample measured with the enzyme-linked immunosorbent assay (ELISA). The cut-off values are shown by red dashed lines, which serve as thresholds for seropositivity. Significant differences in seroreactivity between age groups are presented with red lines and asterisks (Mann–Whitney test, * *p* ≤ 0.05, ** *p* ≤ 0.01).

**Figure 3 viruses-14-02282-f003:**
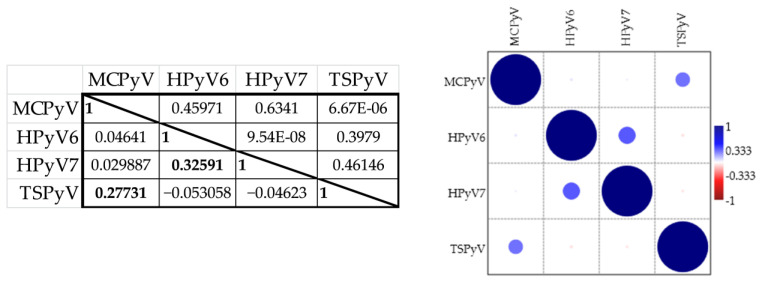
Correlation of seroresponses between polyomaviruses. The left lower triangle of the table shows the correlation coefficients of the OD values from MCPyV, HPyV6, HPyV7 and TSPyV enzyme-linked immunosorbent assays (Spearman’s rank correlation), while the upper right triangle shows the *p*-values. Significant correlation coefficients are presented in bold. The heatmap provides a graphical representation of the correlation coefficients.

**Figure 4 viruses-14-02282-f004:**
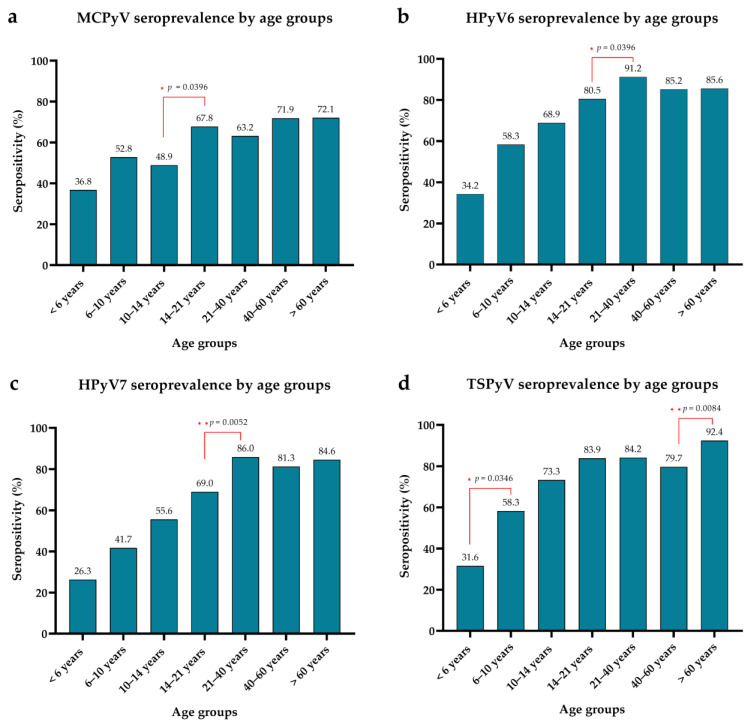
Age-specific seroprevalence of MCPyV (**a**), HPyV6 (**b**), HPyV7 (**c**) and TSPyV (**d**). Significant differences between age groups are presented with red lines and asterisks (Fisher’s exact test, * *p* ≤ 0.05, ** *p* ≤ 0.01).

**Figure 5 viruses-14-02282-f005:**
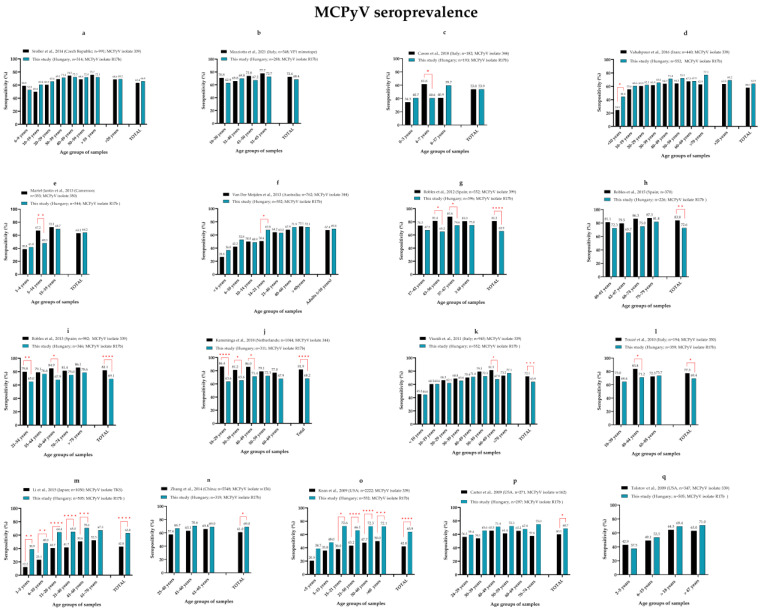
Pairwise comparison of age-specific MCPyV seroprevalence. Each panel represents a comparison between our results and one published study which are the followings: (**a**) [41], (**b**) [16], (**c**) [42], (**d**) [43], (**e**) [44], (**f**) [35] (**g**) [45], (**h**) [40], (**i**) [46], (**j**) [38], (**k**) [47], (**l**) [48], (**m**) [49], (**n**) [50] (**o**) [51], (**p**) [52] and (**q**) [39]. Significant differences between age groups are indicated by red lines and asterisks (Fisher’s exact test, * *p* ≤ 0.05, ** *p* ≤ 0.01, *** *p* ≤ 0.001, **** *p* ≤ 0.0001).

**Figure 6 viruses-14-02282-f006:**
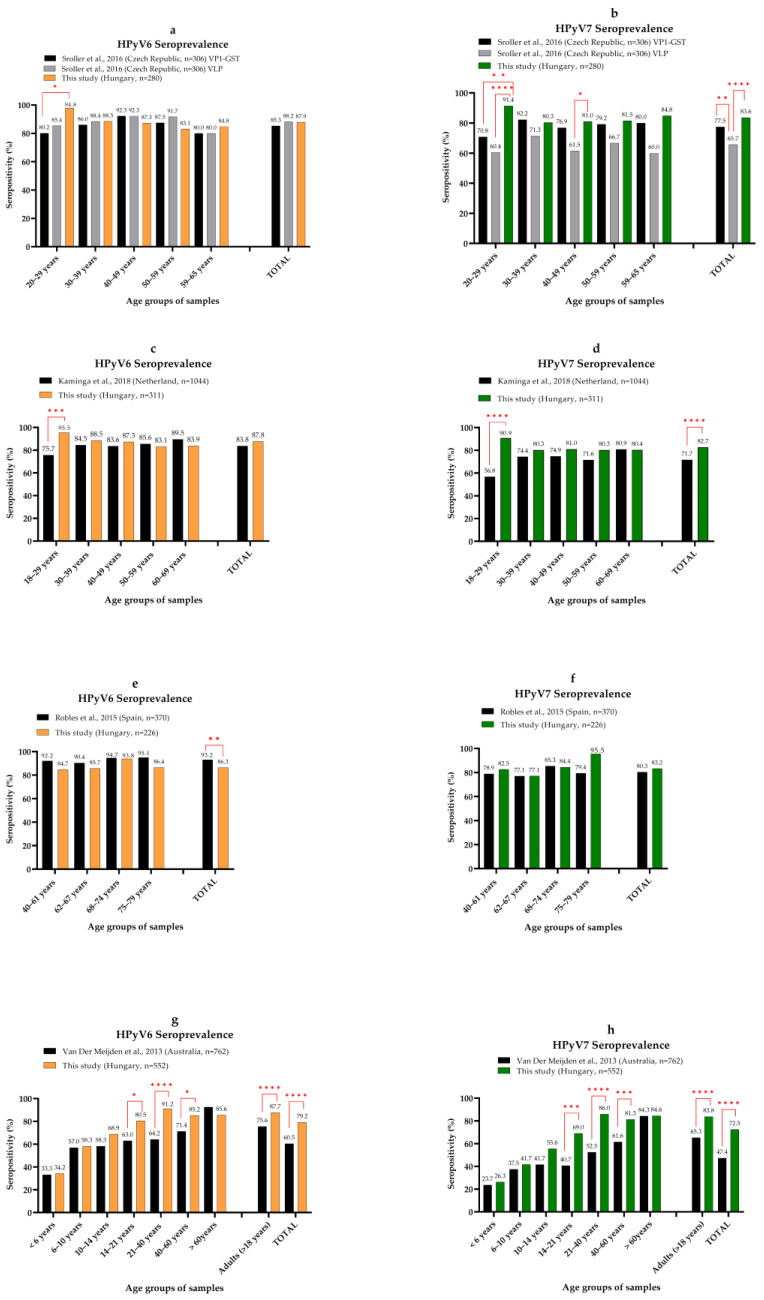
Pairwise comparison of age-specific HPyV6 and HPyV7 seroprevalence. Each panel represents a comparison between our results and one published study which are the the followings: (**a**,**b**) [36], (**c**,**d**) [38], (**e**,**f**) [40] and (**g**,**h**) [35]. Significant differences between age groups are indicated by red lines and asterisks (Fisher’s exact test, * *p* ≤ 0.05, ** *p* ≤ 0.01, *** *p* ≤ 0.001, **** *p* ≤ 0.0001).

**Figure 7 viruses-14-02282-f007:**
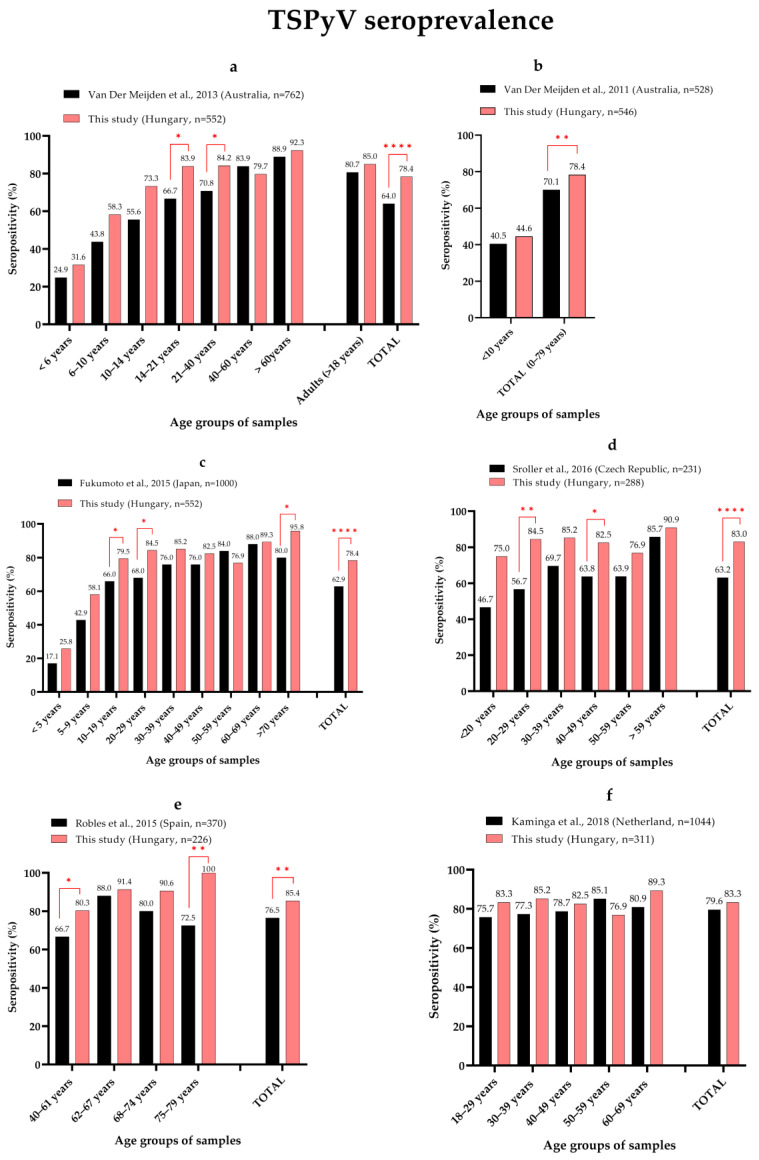
Pairwise comparison of age-specific TSPyV seroprevalence. Each panel represents a comparison between our results and one published study which are the followings: (**a**) [35], (**b**) [53], (**c**) [54], (**d**) [36], (**e**) [40] and (**f**) [38]. Significant differences between age groups are indicated by red lines and asterisks (Fisher’s exact test, * *p* ≤ 0.05, ** *p* ≤ 0.01, **** *p* ≤ 0.0001).

**Table 1 viruses-14-02282-t001:** Characteristics of the study participants by age group.

Age Groups	Number of Samples	Age in Years, Min–Max (Median)	Female/Male
<6 years	38	0.8–5.9 (2.8)	15/23
6–10 years	36	6.1–9.9 (7.6)	19/17
10–14 years	45	10.1–13.9 (11.7)	22/23
14–21 years	87	14–20 (16)	47/40
21–40 years	114	21–39.5 (30.8)	57/57
40–60 years	128	40–59.5 (50)	66/62
>60 years	104	60–85 (69)	57/47
**Total**	**552**	**0.8–85** **(33)**	**283/269**
Adults	359	18–85 (47.3)	187/172
Children	193	0.8–17.9 (11.7)	96/97

**Table 2 viruses-14-02282-t002:** Seroprevalence of MCPyV, HPyV6, HPyV7 and TSPyV by age group and sex.

		Number of Seropositive Samples (Female/Male)		
Age Groups	MCPyV	HPyV6	HPyV7	TSPyV
<6 years	14 (6/8)	13 (6/7)	10 (5/5)	12 (4/8)
6–10 years	19 (12/7)	21 (15/6) *****	15 (10/5)	21 (8/13) *****
10–14 years	22 (14/8)	31 (17/14)	25 (15/10)	33 (15/18)
14–21 years	59 (31/28)	70 (37/33)	60 (36/24)	73 (40/33)
21–40 years	72 (36/36)	104 (51/53)	98 (49/49)	96 (49/47)
40–60 years	92 (52/40)	109 (58/51)	104 (54/50)	102 (52/50)
>60 years	75 (44/31)	89 (50/39)	88 (49/39)	96 (55/41)
**Total**	**353 (195/158) ***	**437 (234/203) ***	**400 (218/182) ***	**433 (223/210)**
Adults	249 (137/112)	315 (166/149)	301 (159/142)	305 (162/143)
Children	104 (58/46)	122 (68/54) *****	99 (59/40) ******	128 (61/67)

Significant differences between groups are indicated with red asterisks (Fisher’s exact test, * *p* ≤ 0.05, ** *p* ≤ 0.01).

## Data Availability

All data are contained within the article.

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
