# Peer review of "Seroprevalence of Four Polyomaviruses Linked to Dermatological Diseases: New Findings and a Comprehensive Analysis"

_viruses, 2022, doi:10.3390/v14102282_

Round 1

Reviewer 1 Report

Date: Sept. 21, 2021

Title: Seroprevalence of four polyomaviruses lined to dermatological disease:  New finding and a comprehensive analysis

Authors: Jeles et al.,

Journal: Viruses

Review Comments:

Two members of the polyomavirus family are known to have pathological characteristics, the primary examples of which are JC virus and BK virus, causing “PML” and “hemorrhagic cystitis and nephritis” respectively. Several other members were also recognized as the disease-causing entities, including, MCPyV, HPyV6, HPyV6 and TSPyV.

The primary goal of this study was to investigate the seroprevalence of the recently discovered disease-causing members, MCPyV, HPyV6, HPyV6 and TSPyV, employing an ELISA assay, developed against the major capsid protein of each virus, and evaluating the results by using various statistical parameters to determine whether there are significant age-related differences both within and among the virus types.

Specific comments:

1.      Grammatical and typing errors could be corrected.

2.      The quality of each VP1 protein used in ELISA for each virus should be evaluated by either Coomassie or silver staining and presented as a separate figure to improve the quality of this manuscript.

3.      The presentation of the Fig. 4 and 5 should be improved. As they stand, the legends of figures are poorly readable.

4.      For Fig. 1a, authors should provide additional explanations why MCPyV seropositivity was not significantly different among the age groups.

Author Response

Response to Reviewer 1 Comments

Point 1.      Grammatical and typing errors could be corrected. 

Response 1: Since we are not native English speaker, high-quality proofreading and language editing was done by Proof-Reading-Service.com.

Point 2.      The quality of each VP1 protein used in ELISA for each virus should be evaluated by either Coomassie or silver staining and presented as a separate figure to improve the quality of this manuscript.

Response 2: Coomassie brilliant blue stained SDS-PAGE photo has been incorporated into the manuscript as Figure 1. As a result of this, all the figures have been renamed.

Point 3.      The presentation of the Fig. 4 and 5 should be improved. As they stand, the legends of figures are poorly readable.

 Response 3: The requirements for the Figures are strict, which was followed and fulfilled. Each figures as a TIF RGB file has the resolution 600 dpi, the size exceeds the minimum expected 1000 pixels, the size of the letters and numbers are correct, the same as used for the other figures (Fig 1 and 3). According  to the instruction, the figures must be inserted into the word text, so without magnification 10% is shown (due to the size of a word page). However, the figures can be enlarged. Since Viruses publishes the articles electronically, images will be able to enlarge to view them at 100% size (it is possible also in the manuscript file). We accept that the letters may not be readable in a 10% view, but all the letters and the numbers are clearly visible at a smaller magnification of even 20%.  Figure 1 and 3 are presented in 14 % of their original size in the manuscript, and these figures are readable.

Point 4.      For Fig. 1a, authors should provide additional explanations why MCPyV seropositivity was not significantly different among the age groups.

  Response 4: We did detect and show significant difference in seropositivity for MCPyV between age groups. As it is detailed in the manuscript, the seropositivity for MCPyV increased significantly with age (Χ2 test for trend; p = 0.0004).  Figure 1a represents seroreactivity and not seropositivity. Seropositivity of MCPyV is shown in the Figure 3a, and statistically significant difference was shown between two age groups (signed also in the figure). Although the age groups which show statistically significant seropositivity rates are varying by the studied viruses, this result for MCPyV is also strengthened by other publications (see figure 5 in the revised manuscript). At the same time, we did not reveal statistically significant difference in seroreactivity (Figure 1a) difference between consecutive age groups, even using the appropriate analysis. In our opinion, it is not evident that differences in seropositivity must be characterized by statistically different seroreactivity. It must be highlighted that due to the new figure as Figure 1, the mentioned Figure 1a has been named to Figure 2a, while Figure 3a has been named to Figure 4a in the revised manusript.

Reviewer 2 Report

In “Seroprevalence of Four Polyomaviruses Linked to Dermatological Diseases: New Findings and a Comprehensive Analysis,” the authors performed ELISA against the VP1 proteins of 4 “cutaneous” polyomaviruses, MCPyV, HPyV6, HPyV7, and TSPyV using the serum of 552 Hungarian individuals. Overall, the data set is comprehensive and largely corroborates what is already known about the seroprevalence of these viruses. The following comments are intended to improve the manuscript:

1)    HPyV6 and HPyV7 are closely related viruses. The VP1 proteins are 69% identical and even more homologous. Thus, the high degree of correlation in the seroresponses to those capsids is somewhat concerning. The authors write that there is evidence that seroresponses to HPyV6 and HPyV7 are specific (data not shown). Could they elaborate more specifically on this data or present it as supplemental data?

2)    The slides comparing seroprevalence between different studies (Fig. 4-6) is difficult to visualize and interpret, particularly since much of it is concordant with what has been published. It may be more helpful to present a table which highlights only the differences between seroprevalence studies. The complete comparisons could be moved to supplemental figures where they would be available for the cognoscenti.

Author Response

Point 1.      HPyV6 and HPyV7 are closely related viruses. The VP1 proteins are 69% identical and even more homologous. Thus, the high degree of correlation in the seroresponses to those capsids is somewhat concerning. The authors write that there is evidence that seroresponses to HPyV6 and HPyV7 are specific (data not shown). Could they elaborate more specifically on this data or present it as supplemental data?

Response 1: The degree of correlation between HPyV6 and HPyV7 is 0,32, which is not high (references are cited to prove it). Other publications cited in the manuscript (ref 35-37) published correlation coefficient for HPyV6 and HPyV7 seroreactivity, and even if the coefficient was higher, 0.433 and 0. 48, respectively, the ELISA method was declaired to be specific enough. As it is written in our manuscript, we proved it by neutralization test as others did.  Some sera chosen for this test were pre-incubated with homolougous or heterolougous antigen, the ELISA measurements were performed after that. The pre-incubation with homolougous antigens resulted in neutralazitaion, hence seronegativity, while the heterolougous antigen did not changed the seroreactivity significantly. These experiments are only for methodology and regularly not published by others.

Point 2.      The slides comparing seroprevalence between different studies (Fig. 4-6) is difficult to visualize and interpret, particularly since much of it is concordant with what has been published. It may be more helpful to present a table which highlights only the differences between seroprevalence studies. The complete comparisons could be moved to supplemental figures where they would be available for the cognoscenti.

Response 2: Summarizing the data comparison in a table would result in loss of valuable results, especially if we highlighted only the differences.  One of the most important results of this comprehensive analysis is that despite the differences in methods, we revealed similarity not only in the numbers of seropositivity, but also in the trends of seropositivity by age. At the same time, even if we determine statistically significant difference in total numbers, data of the same age groups from different studies are not different, revealing similarities. And the reverse was shown also. The not significantly different overall seropositivities showed some significant differences in some groups between the studies. These are explained in the sentences of the manuscript. Trends, whether if statistically significant or not can be visualized by figures more easily than in a table. We think that these figures are valuable and important part of the manuscript, we would not like to delete or published them as supplementary material. The other reviewer also did not suggest it.   In our opinion, table view would be much less clear. The reviewer wrote that these are “concordant with what has been published”, but we did not find any publication presenting the same, comprenehsive analysis. We did not find any publication which analyzed the available data age groups by age groups comparing the publications comprehensively.  The size, resolution have been checked again, it exceeds the requirements: with magnification all the letters, numbers and symbols are readable (incorporation into the word file resulted in 10% appearance of the original size due to the page size, but the figure  can be magnified in the manuscript)

Round 2

Reviewer 1 Report

Date: Oct 10, 2021

Title: Seroprevalence of four polyomaviruses lined to dermatological disease:  New finding and a comprehensive analysis

Authors: Jeles et al.,

Journal: Viruses

Review Comments:

Two members of the polyomavirus family are known to have pathological characteristics, the primary examples of which are JC virus and BK virus, causing “PML” and “hemorrhagic cystitis and nephritis” respectively. Several other members were also recognized as the disease-causing entities, including, MCPyV, HPyV6, HPyV6 and TSPyV.

The primary goal of this study was to investigate the seroprevalence of the recently discovered disease-causing members, MCPyV, HPyV6, HPyV6 and TSPyV, employing an ELISA assay, developed against the major capsid protein of each virus, and evaluating the results by using various statistical parameters to determine whether there are significant age-related differences both within and among the virus types.

Specific comments:

Authors satisfactorily addressed most of the concerns raised by this reviewer except the following question:

1.      The presentation of the Fig. 4 and 5 should be improved. As they stand, the legends of figures are poorly readable.

Authors still did not satisfactorily address this question. Annotations on the figures are not readable without magnification. It is recommended that authors either would drop some of the panels on these figures and state them on the text only or make them visible by adding some more pages to the paper.